# Determining the Optimal Workrate for Cycle Ergometer Verification Phase Testing in Males with Obesity

**DOI:** 10.3390/sports9020030

**Published:** 2021-02-20

**Authors:** Jenny M. Mahoney, Brett R. Baughman, Ailish C. Sheard, Brandon J. Sawyer

**Affiliations:** 1Departments of Biology, Point Loma Nazarene University, San Diego, CA 92106, USA; bsawyer@pointloma.edu; 2Department of Kinesiology, Point Loma Nazarene University, San Diego, CA 92106, USA; bbaughman1215@pointloma.edu; 3School of Kinesiology, Nutrition and Food Sciences, California State University Los Angeles, Los Angeles, CA 90032, USA; asheard@calstatela.edu

**Keywords:** maximal oxygen uptake, true VO_2max_, males with obesity, verification phase, critical power

## Abstract

The aim of the present study was to assess the validity of verification phase (VP) testing and a 3 min all-out test to determine critical power (CP) in males with obesity. Nine young adult males with a body mass index (BMI) ≥ 30 kg·m^−2^ completed a cycle ergometer ramp-style VO_2max_ test, four randomized VP tests at 80, 90, 100, and 105% of maximum wattage attained during the ramp test, and a 3 min all-out test. There was a significant main effect for VO_2max_ across all five tests (*p* = 0.049). Individually, 8 of 9 participants attained a higher VO_2max_ (L/min) during a VP test compared to the ramp test. A trend (*p* = 0.06) was observed for VO_2max_ during the 90% VP test (3.61 ± 0.54 L/min) when compared to the ramp test (3.37 ± 0.39 L/min). A significantly higher VO_2max_ (*p* = 0.016) was found in the VP tests that occurred below 130% of CP wattage (N = 15, VO_2max_ = 3.76 ± 0.52 L/min) compared to those that were above (N = 21, VO_2max_ = 3.36 ± 0.41 L/min). Our findings suggest submaximal VP tests at 90% may elicit the highest VO_2max_ in males with obesity and there may be merit in using % of CP wattage to determine optimal VP intensity.

## 1. Introduction

The most commonly used measurement to assess cardiorespiratory health and fitness is the maximum rate of oxygen consumption (VO_2max_). Although the concept has long been regarded as the gold standard for determining cardiorespiratory fitness [1], there is still much debate on the most reliable method to identify the attainment of a ‘true’ maximal effort [2,3,4]. The plateau phenomenon, described as a leveling off or minimal change in the volume of oxygen consumed (VO_2_) as workrate continues to increase during an exercise test, is the most widely used criterion when determining if a ‘true’ VO_2max_ has been elicited [5]; however, the incidence of a plateau varies from 17–100% due to a wide range of testing protocols, criteria, and means of interpreting data when determining if an individual demonstrated a VO_2_ plateau [4,6,7,8]. Due to the controversy surrounding the plateau phenomenon and the low rate of incidence, secondary criteria have been established to confirm the attainment of VO_2max_. Traditional secondary criteria for determination of a ‘true’ VO_2max_ include threshold values for respiratory exchange ratio (RER), heart rate maximum (HR_max_), and post-exercise blood lactate concentration, yet some authors suggest these criteria also have low reliability and validity [4,5].

A more applicable criterion that can be used for a wide variety of methodologies, protocols, and participants is verification phase (VP) testing. This technique uses a constant workrate bout of exercise following the ramp test to confirm the VO_2max_ achieved during the initial ramp test. The ramp test (or similar graded exercise test) is used to determine a preliminary VO_2max_ and maximal workrate by use of a gradual increase in workrate designed to lead to exhaustion in the recommended time of 8–12 min [9]. By establishing maximal workrate during the ramp test, researchers are then able to choose a verification phase intensity (constant workrate) that falls above critical power (IE: In the severe domain) [10,11] which will lead to exhaustion at VO_2max_. Traditionally a supramaximal rate of the initial maximum workrate is applied [4,6,12,13,14,15,16,17,18,19,20,21,22,23,24,25] but some evidence suggests maximal [10] and submaximal workrates [3,17,21,26,27] are also capable of eliciting a ‘true’ VO_2max_. Verification phase testing eliminates the necessity to rely on primary and secondary criteria to determine if a maximal effort was given; however, no standardized VP testing protocol exists. Research has shown that workrates as low as 80% and as high as 135% of the maximum workrate attained during a ramp test are sufficient to elicit a ‘true’ VO_2max_ [18,27]. Recently, Sawyer et al. [27] concluded that 80% and 90% workrates are capable of eliciting a significantly higher VO_2max_ in young, healthy males compared to supramaximal workrates of 105%, however it is unknown if this would hold true for obese males. It is clear from the current available evidence that the workrates vary between populations (i.e., healthy young adults and adults with obesity) and the application of a standard workrate to confirm the attainment of a VO_2max_ needs to be further explored.

Currently there are a limited number of VP studies that have been published on adults with obesity (body mass index (BMI) ≥ 30 kg·m^−2^) [10,24,25,28]. Due to the limited studies on VP testing in this population, no standard criteria have been implemented into current research. Verification phase testing may be a more accurate way to attain a ‘true’ VO_2max_ in this population, which is an essential measurement for diagnostic, prognostic, and functional information when prescribing exercise [29]. Before this criterion could possibly replace primary and secondary criteria, a standard protocol for the duration and workrates must be obtained. This requires further research on supramaximal, maximal, and submaximal workrates to determine what elicits the highest VO_2max_ in individuals with obesity.

Critical power (CP) has been defined as the highest level of power output in which VO_2_ and lactate can stabilize during exercise and does not lead to a progressive loss of homeostasis [11]. According to Sedgeman et al. [21], CP is useful in detecting the upper limits of verification workrates and durations in relation to ramp tests. Theoretically, power outputs above CP, but below the demarcation of the “extreme domain”, should end in exhaustion at VO_2max_ [11]. Therefore, a VP test that results in too long or too short of a duration and does not attain a VO_2max_ may fall out of this range and the CP test may prove useful in understanding why a ‘true’ max was not elicited. The 3 min all-out test has been well established as an accurate and reliable means for determining CP [30], but its feasibility in males with obesity has not yet been determined. This could be due to previous guidelines including obesity as a risk factor for participating in vigorous exercise [31], which have now been replaced by updated guidelines focused on previous exercise experience and presence of disease.

Therefore, the aim of the present study was to determine the optimal workrate of VP testing in males with obesity. Secondarily, we sought to determine the feasibility of 3 min all-out testing for determining CP in males with obesity.

## 2. Materials and Methods

### 2.1. Subjects

To determine sample size, we used G power [32] along with previously published data [10] and found a sample size of 8 was needed to produce 90% power to detect a significant difference in VO_2max_ between two VP tests. Nine healthy males with obesity between the ages of 18 and 35 volunteered to participate in the six-visit study. See Table 1 for participants’ descriptive characteristics. All participants had a BMI ≥ 30 kg·m^−2^. Participants were between 1 (light activity) and 6 (approximately 3 h of exercise per week) on a self-reported physical activity scale [32]. Participants that answered, “yes” to any questions on the Physical Activity Readiness Questionnaire (PAR-Q) or had any known cardiovascular or respiratory illnesses were excluded from the study [33]. After a verbal and written explanation of the experimental procedure was given, all participants provided written and informed consent. The Institutional Review Board at Point Loma Nazarene University approved all procedures (PLNU IRB ID #1617) and corresponded to the ethical standards of the Declaration of Helsinki.

### 2.2. Experimental Design

In preparation for each test, participants were asked to fast for 4 h before each session and to abstain from alcohol, supplements, caffeine, and strenuous exercise 24 h before each test. If participants did not comply with pre-testing guidelines, the testing sessions were postponed. All six sessions were scheduled at the same time of day (±1 h) and were scheduled a minimum of 48 h apart to ensure each participant was fully recovered between sessions.

Subjects were familiarized with the equipment to prepare for a ramp-style VO_2max_ test on the cycle ergometer (Lode Corival, Groningen, The Netherlands). Ventilation and respiratory gas exchange data were recorded continuously with an automated mixing chamber system (Parvomedics TrueOne 2400, Parvomedics, Sandy, UT, USA) that was calibrated before each testing session or every 4 h, as recommended by the manufacturer. Heart rate was continuously recorded using a Polar hear rate monitor (Polar, Lake Success, NY, USA). Blood lactate was measured after exercise using a Nova Biomedical lactate analyzer (Nova Biomedical Lactate Plus, Waltham, MA, USA).

Once participants were familiarized with the equipment, their anthropometric measurements, including height, weight, and BMI, were recorded and they were prepared for the ramp-style test. Participant’s height, age, weight, sex, and physical activity level (0–10) were inserted into a non-exercising VO_2max_ prediction equation [34]. The estimated VO_2max_ was then used to calculate estimated wattage maximum using the standard ACSM metabolic equation for leg ergometry [35] reworked to solve for wattage, see below:Workrate (W) = subject mass (kg) × (VO_2_ − 7)/1.8(1)

We then used the estimated maximum wattage to determine the participant’s optimal wattage increment for the ramp test to lead to exhaustion in approximately 8–12 min, as suggested by previous research [9]. The ramp test included a 2 min resting phase and a 5 min warm-up at 20 W before power continuously increased (every second) by the calculated W·m^−1^. Participants were verbally encouraged throughout the test to ensure a maximum effort was given. Heart rate was continuously recorded and rating of perceived exertion (RPE) was recorded every minute. Participants pedaled at a self-selected pace between 70 and 90 rpm and the test was terminated when the participant could no longer sustain a pedaling cadence ≥65 rpm. Upon termination of the ramp test, blood lactate levels were measured via a finger prick and recorded. VO_2max_ was calculated by averaging the two highest consecutive 15 s VO_2_ L/min values during the test. If participants failed to complete the test in the set duration, they returned to the laboratory on a separate day and the W·m^−1^ increment was adjusted to elicit a VO_2max_ within the desired time.

Participants returned to the lab a minimum of 48 h following the ramp test to complete the VP tests. The maximum wattage achieved during the ramp test was used to determine the participant’s VP wattage and they were randomly assigned VPs of 1 (80%), 2 (90%), 3 (100%), and 4 (105%). Participants were blinded to which VP they were completing. Verification phase tests included a 2 min resting phase and a 5 min warm-up at 50 W before the test began. Participants were provided verbal encouragement throughout each test and were instructed to pedal between 70 and 90 rpm until volitional exhaustion. The subject’s chosen cadence was recorded during the first VP test and during subsequent VP tests, subjects were required to pedal within a 10 rpm cadence range from their initial VP test. Participants were strongly encouraged to cycle for a minimum duration of 2 min during the VP testing, as recommended by Astorino et al. [6]. Heart rate was continuously recorded throughout the test. Every minute RPE was recorded, and upon termination of the test, blood lactate levels were measured and recorded. VO_2max_ was calculated by averaging the two highest consecutive 15 s VO_2_ L/min values during the test.

During the final visit to the lab, participants performed a 3 min all-out test to determine CP on the cycle ergometer (Monark 839E, Bitz, Germany). Torque for this test was determined by calculating 50% of the difference in power between maximum wattage and the power output at estimated gas exchange threshold during the ramp test [36]. Participants warmed up at 4 N·m^−1^ for 5 min and then began the test at the calculated N·m^−1^ [36]. They were instructed to pedal as fast as possible throughout the test and were provided verbally encouragement. Participants were unaware of the time during exercise to ensure a stabilization of power output near the last 30 s of the test. Critical power was calculated by averaging the power output during the final 30 s of the test.

### 2.3. Statistical Analysis

Statistical analyses were performed using SPSS Software (SPSS 26.0; IBM, Armonk, NY, USA). All data were evaluated for meeting the assumptions of parametric tests used via the Shapiro–Wilk test. Repeated measures analysis of variance was used to assess differences in the outcome measures across the five VO_2max_ tests. Post-hoc tests were run using Bonferroni correction, comparing the means between each pair of VP tests and the ramp test. Independent samples t-tests were used to compare the groups split by CP wattage. Pearson correlations were used to test whether there were significant correlations in VO_2max_, HR_max_, time to exhaustion, and percentage of CP wattage between VP tests. Simple linear regression was used to evaluate the relationship between VP VO_2max_ and % of CP wattage that each VP test occurred at. All effects sizes were calculated as Cohen’s d. An α level of *p* ≤ 0.05 was considered statistically significant.

## 3. Results

Thirteen participants volunteered to complete the study; however, four dropped out during the testing; these four subjects had a similar BMI but were slightly older compared with the nine subjects who completed the study (BMI = 32 ± 1.78, Age = 30.25 ± 2.87). Two participants were unable to complete the ramp tests in a minimum of 8 min, the recommended duration of an exercise test [9,26,37], and were therefore excluded from the study after two attempts. The two other participants decided not to continue the study after two visits to the laboratory due to scheduling constraints. Therefore, nine male participants were included in the data analysis. All data were checked for normality via the Shapiro-Wilk test and all P values were >0.10. The participant characteristics are shown in Table 1. Table 2 depicts the mean and standard deviation of the exercise responses during the ramp and VP tests. Individually, all participants but one attained a higher VO_2max_ (L/min) during a VP test compared to the ramp test (see Figure 1A,B) for VO_2_ tracings for a typical subject). Overall, there was a significant main effect for VO_2max_ between all tests (*p* = 0.049, Figure 2). Bonferroni post-hoc tests showed a trend (*p* = 0.06, effect size = 0.45) for higher VO_2max_ values attained during the 90% VP test (3.61 ± 0.54 L/min) when compared to the ramp test (3.37 ± 0.39 L/min). The VO_2max_ values attained during the 90% VP test (3.61 ± 0.54 L/min) and the 105% VP test (3.41 ± 0.53 L/min) were not statistically significant but showed a small to moderate effect size (*p* = 0.58, effect size = 0.37). Time to exhaustion was not significantly correlated with a higher VO_2max_ (r = 0.32, *p* = 0.061).

There was a main effect for HR_max_ across all tests (*p* = 0.014), but post-hoc testing showed no paired differences (*p* > 0.05). Most participants (seven of nine) achieved a higher HR_max_ during submaximal workrates (80% or 90%) than those attained during the ramp test. Time to exhaustion was not significantly correlated to a higher HR_max_ (r = 0.26, *p* = 0.13).

The CP test was well tolerated in all nine individuals who performed the test and the critical power values calculated were within the expected range for each individual based on their Ramp VO_2max_ test results (mean CP was 73% of mean max wattage on the ramp test; see results in Table 1 and Table 2). Out of the 36 VP tests, only 2 had a wattage that was below CP. The percentage that the wattage of a VP test was above CP was significantly inversely correlated with the percentage that the elicited a higher VO_2max_ during the VP compared to the ramp VO_2max_ (r = −0.39, *p* = 0.018). The percentage that VP wattage was above CP wattage was also significantly inversely correlated with VO_2max_ (r = –0.59, *p* < 0.001). Additionally, when the VP tests were split using a cut-off of above or below 130% of the CP wattage, a significantly higher VO_2max_ (*p* = 0.016) was found in the VP tests that occurred below 130% of CP wattage (N = 15, VO_2max_ = 3.76 ± 0.52) compared to those that were above (N =21, VO_2max_ = 3.36 ± 0.41; see Figure 3).

## 4. Discussion

The present study’s main finding is that VP tests at submaximal workrates, 90% of maximum wattage attained during the ramp test, may elicit the highest VO_2max_ in males with obesity. Our findings suggest that submaximal VP testing on the cycle ergometer may result in the attainment of a higher VO_2max_ compared to the ramp test in this population. Furthermore, the results from the CP testing show that the 3 min all-out test is well tolerated in this population and our data shows VP workrates exceeding 130% of CP may be too high to obtain a ‘true’ VO_2max_ in males with obesity.

To the best of our knowledge, this is the first study to validate the applicability of submaximal workrates during VP testing to elicit a VO_2max_ in males with obesity. There are limited publications that have studied VP testing in men and women with obesity [10,24,25,28], all of which have focused on maximal and supramaximal workrates. Sawyer et al. [10] demonstrated the applicability of maximal (100%) VP testing in adult males (N = 10) and females (N = 9) with obesity (BMI ≥ 30 kg·m^−2^) on the cycle ergometer. While VO_2max_ values did not differ between the ramp and VP tests, most participants (13 of 19) had a VO_2_ value of at least 2% higher in the VP test, further validating the applicability of VP testing in this population. Three of the previous studies [24,25,28] examined supramaximal workrates, two of which found no significant differences in VO_2max_, despite subjects attaining a higher VO_2max_ during VP tests set within 0.5 km·h^−1^ of the maximum workrate of the initial graded exercise test [28] or 105% of maximum wattage [24]. Moreno-Cabanas et al. [25] recently recruited 100 middle-aged metabolic syndrome adults (66 men and 34 women) with a BMI of 32 ± 5 kg·m^−2^ to perform an incremental exercise test on the cycle ergometer. Following the initial test, subjects rested for 15 min then completed a VP test at 110% of maximum wattage. Overall, VO_2max_ was 3% higher (*p* < 0.001) in the VP test compared to the ramp test.

While these four studies have provided greater insight to the ability of individuals with obesity to perform VP tests set at maximal [10] or supramaximal [24,25,28] workrates, none of the studies examined submaximal workrates. Our findings agree with those previously recorded [10,24,25,27,28] in that VP tests may elicit a higher VO_2max_ in these individuals compared to those attained during the ramp test. We have also further validated that individuals with obesity are capable of performing VP testing without complications [10,24,25]. However, contrary to previous findings, the results of the current study suggest that, in this population, a submaximal workrate at 90% of maximum wattage attained during a ramp test may elicit the highest VO_2max_ and workrates greater than 130% of CP (many of the 100 and 105% VP tests) may potentially produce lower VO_2max_ values in obese male participants.

A trend towards a lower VO_2max_ during workrates set at 105% of maximum wattage was seen, indicating that in this population, 105% may be too high of a workrate in the severe exercise domain to elicit a ‘true’ VO_2max_. In all but one subject, each VP test elicited a VO_2max_ higher than that achieved during the ramp test. There is a trend (*p* = 0.06) towards a higher VO_2max_ during the 90% VP compared to VO_2max_ attained during the initial ramp test. Although these findings are not statistically significant, they indicate the necessity for VP testing on the cycle ergometer in this population for determination of a ‘true’ VO_2max_. These findings extend our previous work that showed similar findings in young, healthy, non-obese males [27].

The CP test was well tolerated by all subjects without injury, all subjects were able to complete the 3 min all-out test, and the calculated CP values were within the expected range demonstrating its feasibility in this population. In all cases, except two, the workrate used during VP tests was above CP. The CP attained for that participant was most likely not accurate due to the fact that VO_2_ during all VP tests was still above that elicited during the ramp test in this individual. For the lower workrate VP tests (i.e., below 130% of CP wattage), our findings agree with previous research [11,18,38,39,40,41,42] in that workrates above CP result in attainment of VO_2max_. Conversely, and in agreement with Sawyer et al. [43] and similar to Hill et al. [18] the higher workrate VP tests (i.e., exceeding 130% of CP wattage) may have been too far into the severe exercise domain or even into the extreme domain to elicit a ‘true’ VO_2max_ and HR_max_ in obese men. The use of CP in combination with VP testing may help investigators determine the optimal workrate in relationship to CP to further validate the attainment of a ‘true’ VO_2max_. Our data suggests that VP workrates of ~110% of CP produce superior VO_2max_ values compared to workrates of ~140% of CP in males with obesity (see Figure 2).

### Strengths and Limitations

One strength of this study is that ramp tests were repeated until time to exhaustion and fell within the optimal range of 8–12 min [9], which ensured an accurate ramp VO_2max_ and maximum wattage. Additionally, we were able to confirm that VP workrates fell above CP via use of the 3 min all-out test. Finally, comparing four different VP workrates, varying 80–105% of ramp wattage, enabled us to compare VO_2max_ attainment throughout the spectrum of severe domain wattages (from 112.9 ± 16.2% to 148.4 ± 21.2%). A potential limitation to the study is the 5 min warm-up during VP tests was set at 50 W for each participant, which may have been too low of a workrate before performing severe domain exercise and caused them to fatigue sooner during the VP tests. However, according to ACSM guidelines [35], warm-up intensities for deconditioned individuals should be set around 30% of maximum wattage. The participants in this study warmed-up at ~20% of their maximum wattage, indicating that 50 W may have been an adequate warm-up for most participants, as seen in previous studies with this population [24,25]. Furthermore, average time to exhaustion, even during the 105% VP test, was greater than 2 min, which should be adequate time to achieve VO_2max_ following a warm-up [6]. Additionally, our study was powered to detect the main effect across all five tests but did not include enough subjects to detect multiple individual differences between every combination of comparisons. Even though our study lacked power on that level, our sample size was similar to previous studies [4,17,22,26,44], we achieved statistical significance for our main effect, and noticed a clear trend in the post-hoc testing. Finally, we recognize body composition is a better marker of obesity than BMI, but chose to use BMI due to its widespread use as the primary means of diagnosing obesity in public health. Even though the relationship between BMI and body composition is considered controversial, BMI is still considered a good predictor of clinical outcomes including type 2 diabetes and obesity [45], therefore it was an adequate measurement to determine whether individuals could participate in the study.

## 5. Conclusions

In summary, our study examined supramaximal, maximal, and submaximal workrates to elicit a VO_2max_ in males with obesity. Our results demonstrate a trend towards VP tests at 90% of the workrate attained during the ramp test to confirm the achievement of a ‘true’ maximal effort. Additionally, we demonstrated that the 3 min all-out test is feasible and well tolerated in this population. Our data suggest that workrates set too far above CP (i.e., VP tests at 105% of workrate max) may be above the severe exercise domain and therefore unable elicit a VO_2max_ and HR_max_ in this population. Therefore, we preliminarily recommend VP tests set at a submaximal workrate of 90% maximum wattage attained during the ramp test to elicit VO_2max_ in males with obesity. However, before a standard criterion can be implemented, more research with a larger sample size in both males and females is required on the optimal workrate for VP testing in individuals with obesity.

## Figures and Tables

**Figure 1 sports-09-00030-f001:**
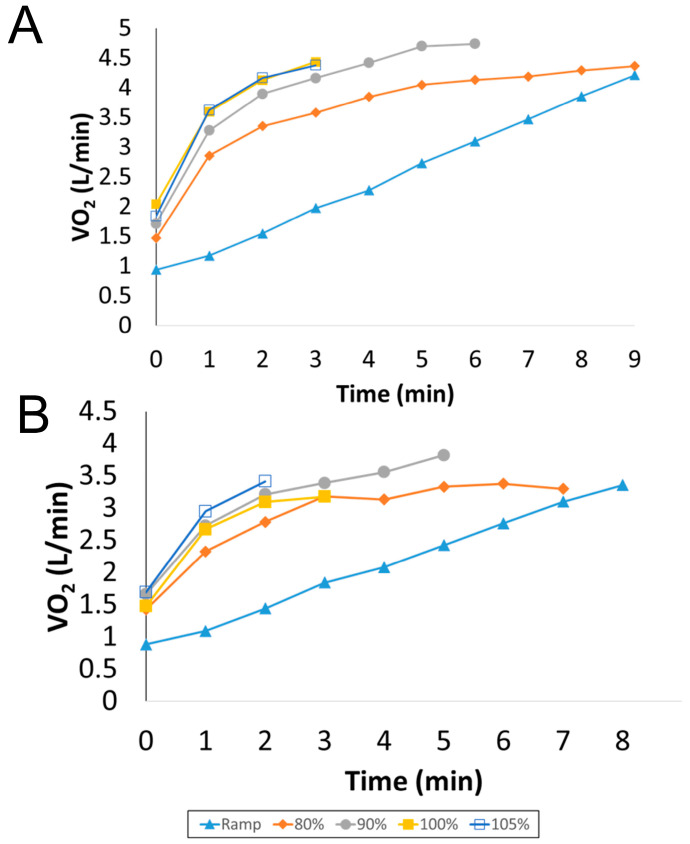
One-minute mean oxygen uptake (L/min) results from two typical participants (**A**,**B**) during ramp and verification phase tests.

**Figure 2 sports-09-00030-f002:**
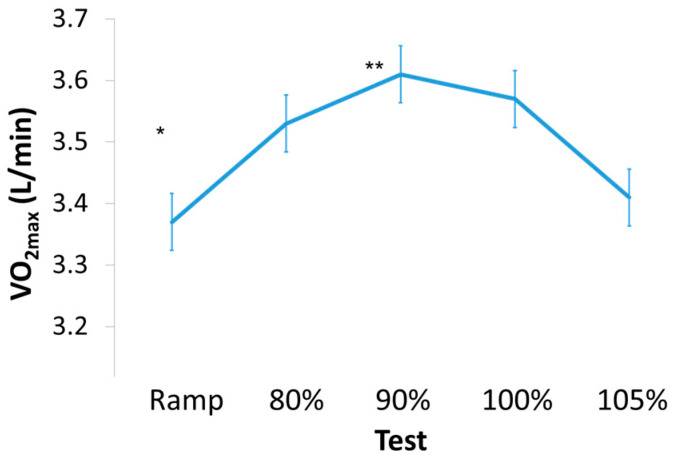
Mean VO_2_ (L/min) values attained during ramp and verification phase tests. * *p* = 0.049, main effect for a significant difference across all tests. ** *p* = 0.06, post-hoc test: 90% VP vs. ramp test. Error bars represent ±1 SD.

**Figure 3 sports-09-00030-f003:**
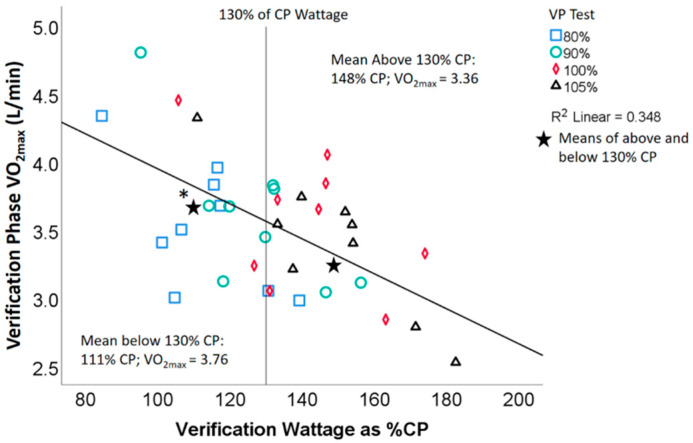
Scatterplot showing the relationship between VP VO_2max_ and verification wattage expressed as a percentage of CP wattage. Linear regression line showing the significant inverse relationship (r = –0.59, *p* < 0.001). Vertical line represents the split between the VP tests occurring above or below 130% of CP wattage. * Mean VO_2max_ below 130% CP wattage was significantly higher than the CP wattage above 130% (*p* = 0.016). VP = verification phase, CP = critical power.

**Table 1 sports-09-00030-t001:** Descriptive statistics of participants.

Variable	Number	Minimum	Maximum	Mean	Standard Deviation
Age (yr)	9	18	34	24	6
Height (cm)	9	170.2	190.5	180.1	6.5
Weight (kg)	9	93.2	149.5	103.0	17.4
BMI (kg m^−2^)	9	30.27	43.40	33.19	4.19
VO_2max_ (ml/kg/min)	9	23.50	46.10	35.13	6.68
Critical Power (W)	9	141	325	205	54

**Table 2 sports-09-00030-t002:** Mean and standard deviation of all outcome variables during ramp and verification tests.

Test	VO_2_ (L/min)	HR_max_ (bpm)	RER	BL (mM)	Peak Wattage	Time (s)	RPE	% of CP Wattage
Ramp	3.37 ± 0.39	175.4 ± 12.5	1.26 ± 0.08	12.3 ± 2.4	281 ± 35	497 ± 24	17 ± 4	–
80%	3.53 ± 0.47	177.1 ± 15.8	1.25 ± 0.06	13.8 ± 2.7	224 ± 28	418 ± 153	19 ± 1	113 ± 16
90%	3.61 ± 0.54	176.2 ± 14.0	1.25 ± 0.08	13.6 ± 3.1	253 ± 32	282 ± 56	17 ± 3	127 ± 18
100%	3.57 ± 0.51	176.0 ± 13.1	1.32 ± 0.10	12.7 ± 2.4	281 ± 35	203 ± 27	17 ± 2	141 ± 20
105%	3.41 ± 0.53	169.8 ± 17.1	1.32 ± 0.05	12.3 ± 2.1	295 ± 37	167 ± 40	17 ± 3	148 ± 21

HR_max_ = heart rate maximum (beats per minute), RER = respiratory exchange ratio, BL = blood lactate, RPE = rating of perceived exertion, CP = critical power.

## Data Availability

The data present in this study are available upon request from the corresponding author. The data are not publicly available due to privacy restrictions.

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
