# Peer review of "Determining the Optimal Workrate for Cycle Ergometer Verification Phase Testing in Males with Obesity"

_sports, 2021, doi:10.3390/sports9020030_

Round 1
Reviewer 1 Report
This study aimed to determine the validity of verification phase testing and a 3-minute all-out test to determine critical power in nine males with obesity.
The study has merit and is well written.
The authors should discuss in more detail about the limitations of the results' statistical significance.
statistical The small size sample and the "poor" statistical "trends" require caution before the implementation of the results and the conclusion about the optimal workrate in this population.
Reviewer 2 Report
Does the introduction provide sufficient background and include all relevant references?
There is no adequate introduction to the problems that the test may have in order to be feasible for obese men.
One of the objectives of this study is to study the feasibility of this test in obese men. This objective is not correctly developed throughout the manuscript, how was viability evaluated, what criteria were used to determine whether the test is viable or not. According to your data in your article, 13 people started and 9 finished, which means that 30% of the participants did not finish the test. What characteristics did the participants who did not finish the test have? If one of the objectives of the study is to see the feasibility of this test, the characteristics of the participants who did not finish must be indicated. It is essential that this appears in the results section and is discussed in depth.
Is the research design appropriate? / Are the methods adequately described?
No information is specified about what type of distribution the data included in the statistical analyses followed. What test was used to determine if the data followed a normal distribution?
Linear regression is included in figure 3, in the section on statistical analysis, there is no mention of this statistical test.
Are the results clearly presented?
Characterization data of the participants who did not complete the test are missing.
Are the conclusions supported by the results?
No conclusion on the feasibility of the test is included.
Reviewer 3 Report
Since the primary method of obesity treatment is lifestyle modification based on a negative energy balance, research aimed at optimizing the parameters of training protocols is of particular value. In their work, Mahoney et al. assessed the validity of verification phase (VP) testing and a 3-minute all-out test to determine critical power (CP) in 9 young males with obesity. They found that in the studied group, submaximal VP tests at 90% may elicit the highest VO2max, and therefore they conclude that using % of CP wattage can be useful to determine optimal VP intensity. The study refers to an important issue, the protocol is clear, and a power calculation supports the number of participants. However, I would like to draw the authors' attention to the one significant limitation that has not been mentioned in the "Strength and limitation" section. The exercise capacity is related to body composition (lean body mass). Even though BMI is commonly used to diagnose obesity, it does not refer to body composition. Therefore the study would benefit if the body composition is assessed in the studied group. Alternatively, the study could include a mixed population of men and women (since there are significant gender differences in body composition).
Reviewer 4 Report
Comments and Suggestions are attached

Round 2
Reviewer 4 Report
Thank you.